# Highly-Tunable Crystal Structure and Physical Properties in FeSe-Based Superconductors

**Kaiyao Zhou [1,2], Junjie Wang [1,2], Yanpeng Song [1,2], Liwei Guo [1] and Jian-gang Guo [1,3,*]**

[1] Beijing National Laboratory for Condensed Matter Physics, Institute of Physics, Chinese Academy of Sciences, Beijing 100190, China; kaiyaozhou@hotmail.com (K.Z.); wangjunjie1219@hotmail.com (J.W.); syp2226340310@hotmail.com (Y.S.); lwguo@aphy.iphy.ac.cn (L.G.)

[2] University of Chinese Academy of Sciences, Beijing 100049, China

[3] Songshan Lake Materials Laboratory, Dongguan, Guangdong 523808, China

* Correspondence: jgguo@iphy.ac.cn

**Abstract:** Here, crystal structure, electronic structure, chemical substitution, pressure-dependent superconductivity, and thickness-dependent properties in FeSe-based superconductors are systemically reviewed. First, the superconductivity versus chemical substitution is reviewed, where the doping at Fe or Se sites induces different effects on the superconducting critical temperature ($T_c$). Meanwhile, the application of high pressure is extremely effective in enhancing $T_c$ and simultaneously induces magnetism. Second, the intercalated-FeSe superconductors exhibit higher $T_c$ from 30 to 46 K. Such an enhancement is mainly caused by the charge transfer from the intercalated organic and inorganic layer. Finally, the highest $T_c$ emerging in single-unit-cell FeSe on the $SrTiO_3$ substrate is discussed, where electron-phonon coupling between FeSe and the substrate could enhance $T_c$ to as high as 65 K or 100 K. The step-wise increment of $T_c$ indicates that the synergic effect of carrier doping and electron-phonon coupling plays a critical role in tuning the electronic structure and superconductivity in FeSe-based superconductors.

**Keywords:** FeSe; superconductivity; high pressure; chemical intercalation; interfacial coupling

## 1. Introduction

After discovering FeAs-based superconductors in 2008 [1–4], a structurally-simple binary FeSe superconductor with a superconducting critical temperature ($T_c$) of 8 K was quickly established by Hsu et al. [5] This brought researchers into a fresh new field of iron-based superconductors. The binary compound is composed of a neutral FeSe layer that is composed of a $FeSe_4$ tetrahedra along the *c*-axis, and thus the interlayer coupling is weak van der Waals forces. It was found, initially, that the chemical substitution of Se by S and Te mildly enhanced the $T_c$ to 10–15 K. However, tiny amounts of doping (3%) in the Fe site by Cu and Co strikingly suppress the $T_c$. Such features are totally different from those of FeAs-based superconductors, where the Co/Ni doping effectively induces superconductivity (SC) accompanying many properties like linear-T resistivity, a quantum critical point, and magnetic fluctuations [6–8]. Furthermore, there is no long-range magnetic order in FeSe, and the interplay between magnetism and SC is lacking, thus its superconducting mechanism seems to be simpler than that of FeAs-based superconductors. On the other hand, it is hard to grow large-sized single crystals of FeSe by the conventional flux-method because superconducting FeSe is a metastable phase that only exists in the temperature range of 673 K−973 K. The intrinsic properties like gap symmetry and the Fermi surface are hence inaccessible. These disadvantages, to some extent, hinder the investigations of FeSe-based superconductors.

Later, a technique of growing high-quality FeSe single crystals was found, and multiple characterizations like magneto-transport, in situ pressure, and inelastic neutron scattering revealed inherent properties for SC. More importantly, in 2010, the intercalation between the FeSe layer by large-sized alkali metals was successfully realized, forming a series of superconductors $A_x Fe_{2-y}Se_2$ (A = K, Cs, Rb, Tl) with a $T_c$ of 30 K. This discovery opened a broader field in the physics and chemistry community. Subsequent reports of alkali metal, and ammonia co-intercalated FeSe, inorganic molecule intercalation, and monolayer FeSe on a $SrTiO_3$ substrate continuously enhanced the $T_c$ up to 65 K or even 100 K, making the FeSe-based compound one of the most fascinating fields in the search for a high $T_c$ superconductor. Apart from the enhancement of $T_c$, the angle resolve photoemission spectrum (ARPES) revealed that the Fermi surface in such high $T_c$ superconductors share a common feature (i.e., only two electron pockets surviving at the M point). A unique Fermi surface possibly implies a different pairing mechanism for Cooper pairs rather than the Fermi nesting scenario between electron-hole pockets in the well-documented FeAs-based superconductor [9,10].

In the last 11 years, FeSe-based superconductors have exhibited huge advancements and prosperous developments, and we feel that it is necessary to summarize the significant progress made at each important step. In this paper, we will review the crystal structure, the electronic structure, and the superconducting phase diagram in diverse FeSe-based superconductors. The evolutions of SC in binary FeSe under chemical substitution and physical pressure are discussed in first part. Second, the FeSe-based intercalates through inorganic and organic molecules are summarized, where both the interlayer spacing between FeSe layers and the charge transfer should determine the $T_c$. Finally, the substrate effect, the doping carrier, and the interfacial coupling of high-$T_c$ SC in thin FeSe film are discussed. We hope that this review can deepen the understanding of structural evolution and superconductivity in FeSe-based superconductors to motivate the exploration of new superconducting materials with higher $T_c$.

## 2. Superconductivity in FeSe

### 2.1. Structure and Stoichiometric of $Fe_{1+x}Se$

Binary FeSe with $T_c$ = 8 K was discovered in 2008. It is regarded as a less-toxic superconductor when compared with the FeAs-based superconductors. The structure of FeSe is the PbO-type, which is composed of FeSe layers that are formed by edge-shared $FeSe_4$ tetrahedra. The onset of $T_c$ is 8.5 K. Rietveld refinements of powder synchrotron diffraction (BL12B2 beamline in Spring8) yield $a = 0.37693(1)$ nm and $c = 0.54861(2)$ nm for $FeSe_{0.82}$, and $a = 0.37676(2)$ nm and $c = 0.54847(1)$ nm for $FeSe_{0.88}$. By cooling below the phase transition temperature ($T_s$) of 90 K, the (220) peak of PXRD splits into (200) and (020), indicating a structural transition from tetragonal $P4/nmm$ to triclinic $P$-1, as shown in Figure 1. After that, Margadonna et al. carefully resolved the low-temperature crystal structures for $FeSe_{0.92}$ [11]. Both high-resolution synchrotron x-ray diffraction and powder neutron diffraction (NPD) were carried out at different temperatures. They found that it indeed experienced a structural transition below the $T_s$, but the low-temperature phase showed an orthorhombic phase (space group $Cmma$), rather than triclinic symmetry, as seen in Figure 2. This transition was later called a nematic transition, which possibly correlates with the SC evidenced by multiple characterizations [12,13].

Pomjakushina et al. used two different methods to synthesize the $FeSe_{1-x}$ powder [14], and found that the superconducting phase existed in a very narrow range of Se content ($FeSe_{0.974\pm0.005}$). Williams et al. carefully identified the role of Se-deficiency on tuning the structural transition and SC [15]. They proved experimentally that the correct formula of superconducting samples was $FeSe_{0.99}$. The Se vacancies may not be necessary for inducing SC. They also determined that the SC was extremely sensitive to the content and disorder of Fe. After introducing 3% excess Fe, the SC was totally suppressed. Additionally, the magnetic order was not observed down to 5 K, in contrast to the antiferromagnetic (AFM) ground state in the FeAs-based superconductors. The $Fe_{1.01}Se$ compound exhibits the superstructure below the $T_s$ in the form of a $\sqrt{2} \times \sqrt{2} \times 1$ supercell, consistent with

the previous report [16]. Moreover, in the compound $Fe_{1.03}Se$, the structural transition disappears, which means that excess Fe can break the long-range coherence of the structural unit. Recently, more details on the superstructure and SC were revealed in vacancy-ordered $Fe_{1-x}Se$. Three kinds of superstructures, $\sqrt{2} \times \sqrt{2} \times 1$, $\sqrt{5} \times \sqrt{5} \times 1$, and $\sqrt{10} \times \sqrt{10} \times 1$, were identified in $Fe_3Se_4$, $Fe_4Se_5$, and superconducting $Fe_9Se_{10}$, respectively [17]. Among them, the $Fe_4Se_5$ compound is an AFM insulator with a small effective magnetic moment (0.003 $\mu_B$). These authors claimed that the Fe vacancies could be effectively tuned, and increasing the content of Fe gradually led to the so-called parent compound $Fe_4Se_5$ into a superconducting state. Such behavior is analogous to the superconducting phase diagram in cuprates, where the parent compounds are AFM insulators and excess hole or electron doping can induce SC [18].

Regarding the single crystal of FeSe, Mok et al. could grow it through the high-temperature method using flux KCl [19]. As shown in Figure 3a, single crystals 2–3 mm in width and 0.1–0.3 mm in thickness were obtained. Only small amounts of the hexagonal FeSe impurity phase was detected according to the x-ray diffraction analysis. This work further proved that post-annealing at 673 K was useful to improve the quality of single crystals. Chareev et al. successfully grew high-quality FeSe single crystals by the chemical vapor deposition method. The atomic ratio of Fe:Se = 1:0.96 ± 0.02 and the maximal size was $4 \times 4 \times 0.1$ mm$^3$. The layered feature can clearly be observed in the Figure 3b. The $T_c$ $^{onset}$ of this crystal was 9.4 K [20]. Following the improvement of single crystals, several groups quickly determined the intrinsic SC of FeSe. Lei et al. presented the nearly isotropic upper critical field [$H_{c2}(0)$] with magnetic field parallel and perpendicular to (101) face (see Figure 4a,b), and deduced a large Ginzburg-Landau parameter of $\kappa \sim 72.3(2)$ [21]. The critical current density ($J_c$) was determined to be $2.2 \times 10^4$ A/cm$^2$ by the Bean model. Furthermore, these authors analyzed the Hall effect of FeSe and confirmed that the dominant carriers were hole-type at high magnetic fields. The electron-type carriers made a larger contribution at low magnetic field and low temperature, as shown in Figure 4c [22]. The symmetry of the superconducting energy gap was investigated by Lin et al. who demonstrated the coexistence of the isotropic *s*-wave and extended anisotropic *s*-wave gap with the magnitude of 1.33 meV and 1.13 meV in FeSe [23]. A high electron-boson coupling constant, 1.55, implies that the electron is not only coupled with phonons, but also with other "glues" associated with spin fluctuations. Subsequently, Abdel-Hafiez et al. measured the temperature-dependent London penetration depth, which was fitted by either a two s-wave-like model or a single anisotropic gap [24]. They claimed that the superconducting energy gap is nodeless. The fitting curves of a two-gap model with anisotropic *s*-wave and *d*-wave models are shown in Figure 4d.

Density functional calculations demonstrated that the electronic structure of FeSe was very similar to those of the FeAs-based superconductors [25]. In the first Brillouin zone, there are heavy hole-cylinders and lighter electron-pockets near the zone center (Γ-point) and corner (M-point), respectively, implying typical spin-density wave instability due to Fermi surface nesting. In 2014, Shimojima et al. investigated the temperature-dependent band dispersion of Fe $3d_{xz}$ and $3d_{yz}$ orbitals at the M point from 50 K to 110 K. The results showed that the removal of degeneracy in the $3d_{xz}/3d_{yz}$ bands occurred at temperatures close to the $T_s$, which may be the orbital origin of structural transition [26]. Later, Zhang et al. traced the temperature dependence of band dispersion at the M point and the Γ point. They surprisingly found that the band splitting at the Γ point was not closely related to the structural transition, which may be controlled by magnetic frustration [27]. A high-resolution laser-based ARPES on FeSe recently uncovered a highly anisotropic Fermi surface around the Γ point. This is possibly related to the low-temperature orthorhombic phase (i.e., nematic state), because the splitting of $3d_{xz}/3d_{yz}$ bands is possibly due to anisotropic electron hopping. Moreover, the observation of an extremely anisotropic superconducting gap with two-fold symmetry was also associated with nematicity. Fitting the gap scale against momentum suggests a possible combination of extended *s*-wave and *d*-wave gap symmetry [28,29]. It can be found that many characterizations, at least, confirmed that the superconducting energy gap of FeSe was of the anisotropic type.

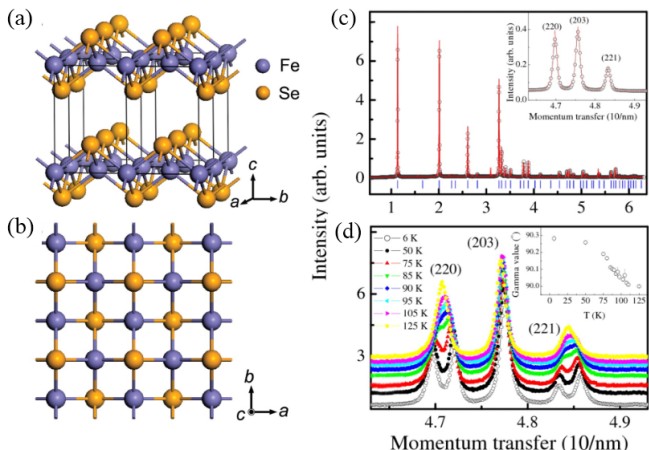

**Figure 1.** (**a**,**b**) Crystal structure of FeSe. (**c**) Observed (open black circle) and calculated (red solid line) powder diffraction intensities of FeSe$_{0.88}$ at 300 K using space group *P4/nmm*. (**d**) The temperature dependence of the $\gamma$-angle fitted with *P-1* symmetry. The single peak of the (2, 2, 0), (2, 0, 3), (2, 2, 1) reflection is seen in (**c**), splitting of the diffraction peaks into two peaks was observed (Figure reprinted from Hsu, F.C. et al. *Proc. Natl. Acad. Sci. U. S. A.* **2008**, 105, 14262–14264. Copyright 2008 by the National Academy of Sciences of the USA).

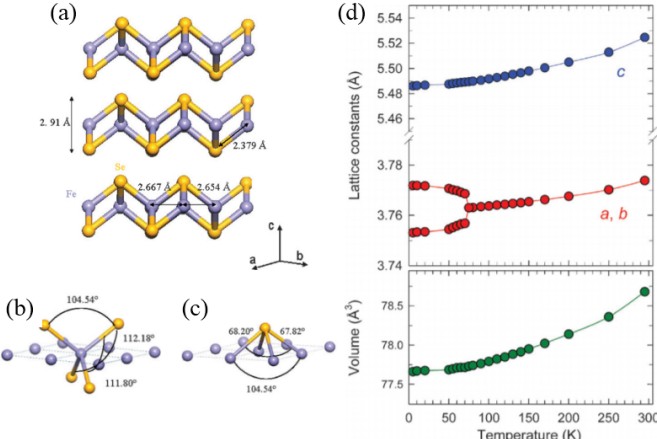

**Figure 2.** (**a**) Schematic diagram of the low-temperature orthorhombic structure of FeSe$_{0.92}$. (**b**,**c**) Geometry of the FeSe$_4$ tetrahedra and the SeFe$_4$ pyramids with three distinct Se–Fe–Se and Fe–Se–Fe bond angles, respectively. (**d**) Temperature dependent lattice constants of FeSe. (Figure reprinted from Margadonna, S. et al. *Chem. Commun. (Cambridge, U. K.).* **2008**, 5607–5609. Copyright 2008 by the Royal Society of Chemistry).

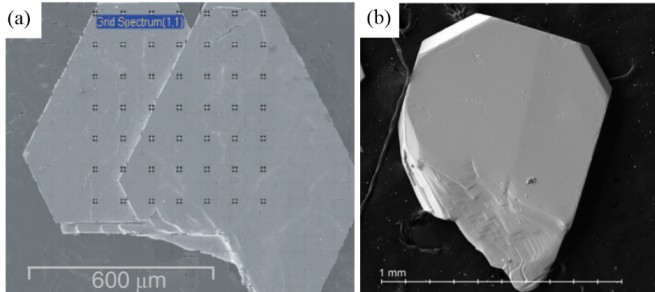

**Figure 3.** (**a**) Scanning electron micrographs of the as-grown FeSe$_{0.9}$ single crystal 2–3 mm in width and 50–60 μm in thickness by the high temperature flux-method. (**b**) The FeSe$_{0.96}$S$_{0.04}$ single crystal was obtained by chemical vapor deposition (Figure 3a reprinted from Vedeneev S.I. et al. *Phys. Rev. B* **2013**, 87, 134512. Copyright 2013 by American Physical Society. Figure 3b reprinted from Chareev, D. et al. *Cryst. Eng. Comm.* **2013**, 15, 1989–1993. Copyright 2013 by the Royal Society of Chemistry).

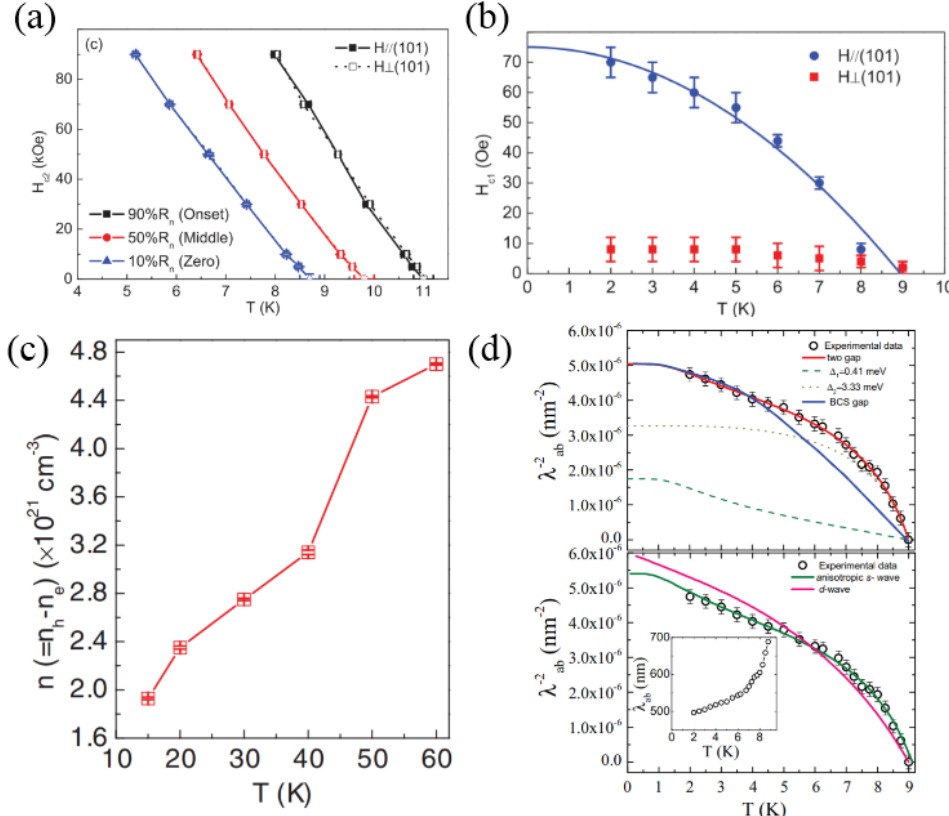

**Figure 4.** (**a**) Determination of the upper critical field $H_{c2}(0)$ from the temperature dependence of the resistivity ρ(T) of *β*-FeSe single crystals for H//(101) and H⊥(101) directions. (**b**) Temperature-dependence of $H_{c1}$ for both H//(101) and H⊥(101) directions. The solid blue line was the fitting curve using $H_{c1}(T) = H_{c1}(0)[1 − (T/T_c)^2]$ for H//(101). (**c**) Temperature dependence of carrier density n ($n_h − n_e$) of the β-FeSe crystal. (**d**) Fitted curve of two-gap model with anisotropic *s*-wave and *d*-wave models (Figure 4a,b reprinted from Lei H.C. et al. *Phys. Rev. B* **2011**, *84*, 014520. Copyright 2011 by American Physical Society. Figure 4c reprinted from Lei H.C. et al. *Phys. Rev. B* **2012**, 85, 094515. Copyright 2012 by American Physical Society. Figure 4d reprinted from Hafiez, M.A. et al. *Phys. Rev. B* **2013**, 88, 174512. Copyright 2013 by American Physical Society].

## 2.2. FeSe under Pressure and Chemical Substitution

The application of high pressure on FeSe exhibits a surprising effect on tuning SC and magnetism. Mizuguchi et al. first enhanced the $T_c$ of polycrystalline FeSe to 13.5 K under 1.48 GPa with a positive pressure-coefficient of 9.1 K/GPa [30]. Linear-extrapolated $H_{c2}(0)$ was increased from 37 T to 72 T, implying the unconventional feature of SC. Millican et al. applied 0.6 GPa hydrostatic pressure to FeSe, and the structural parameters deduced from the high-resolution NPD pattern suggested that the phase transition was suppressed under pressure. A small bulk modulus, 31 GPa, suggested that FeSe is a soft material that can be effectively pressurized [31]. Garbarino et al. observed a new orthorhombic phase at a higher $T_c$ of 34 K under 22 GPa [32]. Imai et al. performed the $^{77}$Se nuclear magnetic resonance (NMR) measurement at 2.2 GPa and ascribed the strongly-enhanced AFM spin fluctuations to be the origin of the enhancement of $T_c$ [33]. This picture was supported by a zero-field muon spin rotation (µSR) measurement, from which the SC and AFM order were simultaneously stabilized in the low pressure range (see Figure 5a) [34]. Meanwhile, a very small magnetic moment, 0.2 $µ_B$, was obtained at 2.4 GPa. In an extended pressure range, a clear dome-shaped superconducting phase diagram as a function of pressure was established, where the maximal $T_c$ of 36.7 K showed up at 8.9 GPa. Above 12 GPa, the tetragonal FeSe changed into hexagonal FeSe (NiAs-type), which is a non-superconducting phase (Figure 5b) [35]. The in situ Mössbauer spectrum under pressure revealed

that there was no long-range magnetic ordering in the whole pressure range. This observation is in stark contrast to the results of the NMR and *μ*SR experiments.

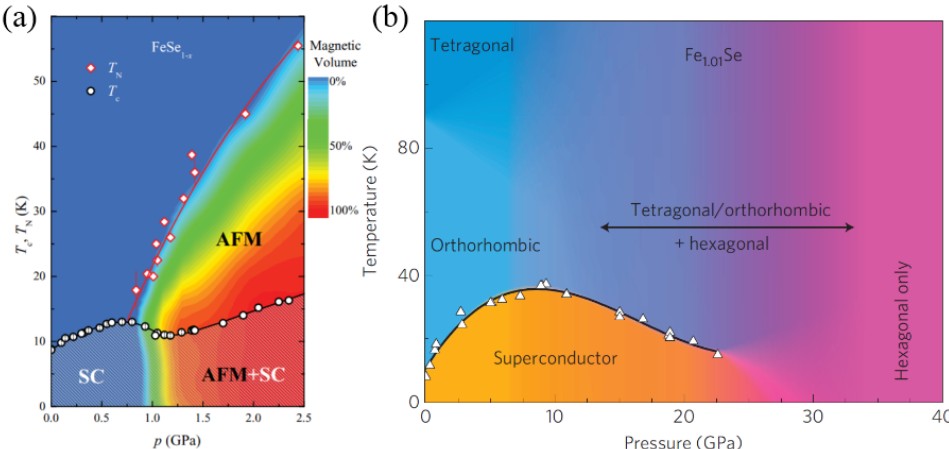

**Figure 5.** Electronic phase diagram of FeSe$_{1-x}$ under pressure. (**a**) Pressure dependence of the $T_c$, $T_N$, and the superconducting and magnetic volume fractions of FeSe$_{1-x}$. (**b**) Electronic phase diagram of Fe$_{1.01}$Se under a pressure range from 0 to 40 GPa. The maximum $T_c$ observed was 36.7 K at 8.9 GPa (Figure 5a reprinted from Bendele M. et al. *Phys. Rev. B* **2012**, 85, 064517. Copyright 2012 by American Physical Society. Figure 5b reprinted from Medvedev S. et al. *Nat. Mater.* **2009**, 8, 630–633. Copyright 2009 by Macmillan Publishers Limited).

To clarify this, single crystals of FeSe have been used in in-situ high-pressure experiments. In 2015, Terashima et al. applied the high pressure of 2.72 GPa on FeSe single crystals and measured the resistance and ac magnetic susceptibility [36]. They confirmed that $T_c$ generally increased under pressure, but dropped to a minimum $T_c$ at ~1.2 GPa, which was consistent with the pressure-induced antiferromagnetic phase transition [37]. At pressures above 5 GPa, Sun et al. observed a sudden enhancement of $T_c$ and suppression of $T_N$, indicating competition between SC and magnetic order (see Figure 6). Furthermore, an interesting linear-T resistivity was observed at higher pressure, which can be considered to be a signature of a quantum critical point [38]. Application of higher pressure would destabilize the AFM state and thus enhance the spin fluctuations, to some extent, which is similar to the mechanism enhancing $T_c$ in FeAs-based superconductors. At the optimum $T_c$, the electrical transport behaviors are dominated by holes, which are enhanced near optimal pressure, implying a Fermi surface reconstruction due to AF ordering [39].

According to the reports by Yeh et al. Te-substitution of Se sites resulted in lattice expansion because of the larger ionic radius of Te, as shown in Figure 7 [40,41]. It also can enhance onset of superconductivity ($T_c^{onset}$) to 15.2 K and $H_{c2}(0)$ to 28.8 T as half of the Se was substituted. Later, a high-quality single crystal of Fe$_{1+y}$Se$_{1-x}$Te$_x$ was grown from the melt [42]. It showed that the single-phase can only exist in the Te-rich side. Bulk SC at 14 K in Fe$_{1+y}$Se$_{0.5}$Te$_{0.5}$ was confirmed simultaneously by resistivity, magnetic susceptibility, and heat capacity. Post-annealing on Te-rich samples could improve the filamentary SC into bulk SC [43–46]. Meanwhile, the excess Fe at the interstitial site not only suppresses SC, but also induces weakly-localized states. From the neutron pair density function analysis, Se and Te do not actually occupy the same crystallographic site, leading to a local symmetry around Fe that is lower than the average crystal symmetry (*P4/nmm*) in Fe$_{1+y}$Se$_{1-x}$Te$_x$ solid-solutions [47]. A comprehensive phase diagram of the SC and the structure of Fe$_{1+y}$Se$_{1-x}$Te$_x$ were constructed by Liu et al. where bulk SC emerged in the range of 50% to 70% Te doping [48]. The in-plane magnetic wave vector in FeTe for the AFM order is ($\pi$, 0); as the content of Se increases, the magnetic fluctuations gradually evolve into the ($\pi$, $\pi$) type associated with the scenario of Fermi surface nesting. This finding first revealed that both FeSe- and FeAs-based superconductors may share a similar mechanism for SC, assuming that the spin fluctuation picture is applicable.

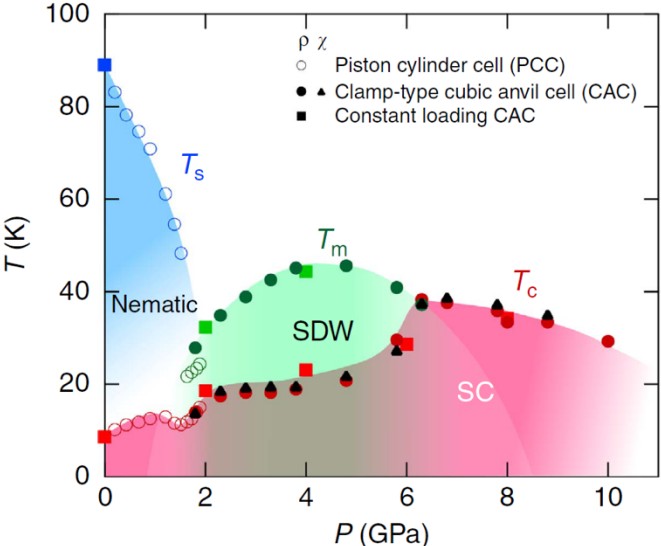

**Figure 6.** Superconducting phase diagram of bulk FeSe under pressure (Figure reprinted from Sun, J.P. et al. *Nat. Commun.* **2016**, *7*, 12146. Copyright 2016 by Macmillan Publishers Limited).

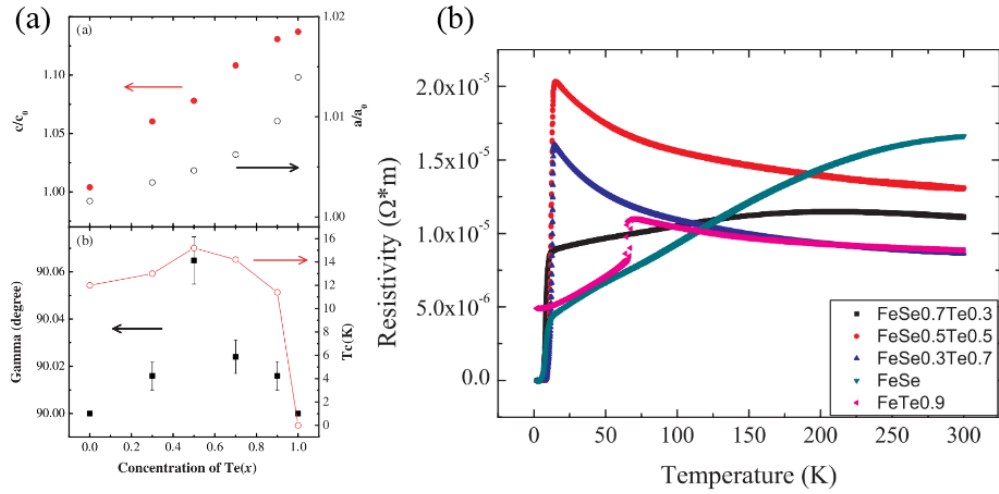

**Figure 7.** (**a**) Te-doping dependence of structural evolution and $T_c$ of $FeSe_{1-x}Te_x$. (**b**) Temperature dependence of $FeSe_{1-x}Te_x$ resistance under zero magnetism (Figure reprinted from Yeh, K.W. et al. *J. Phys. Soc. Jpn.* **2008**, *77*, 19–22. Copyright 2008 by JPS).

Regarding S-doped FeSe, the soluble limit of S is smaller than 20%, according to a report from Mizuguchi et al. [49] Additionally, Abdel-Hafiez et al. studied the superconducting properties of high-quality single crystals of $FeSe_{1-x}S_x$ ($x = 0$, 0.04, 0.09, and 0.11). As the S concentration increases, the $T_c$ determined from the onset of the diamagnetic signal increases from ~8.5 K to 10.7 K at $x = 0.11$. The normalized specific heat jump for $FeSe_{1-x}S_x$ was significantly larger than the limit (1.43) of the BCS model, which is a signature of strong-coupling SC [50]. By using ARPES, Watson et al. observed a smaller splitting of the $3d_{xz}$ and $3d_{yz}$ bands and a weaker anisotropy of the Fermi surface in the $FeSe_{1-x}S_x$ [51]. This means that isovalent S-substitution reduces the orbital ordering, correspondingly suppressing the structural transition temperature from 87 K to 58 K. Matsuura et al. investigated the pressure-dependent phase diagram of $FeSe_{1-x}S_x$, as shown in Figure 8 [52]. As S-content increased, the structural transition was quickly suppressed. Simultaneously, the pressure-induced SDW phase zone was gradually narrowed and the maximal $T_c$ in each composition also decreased. This becomes a single dome-like SC in $FeSe_{0.83}S_{0.17}$, where the highest $T_c$ was 30 K.

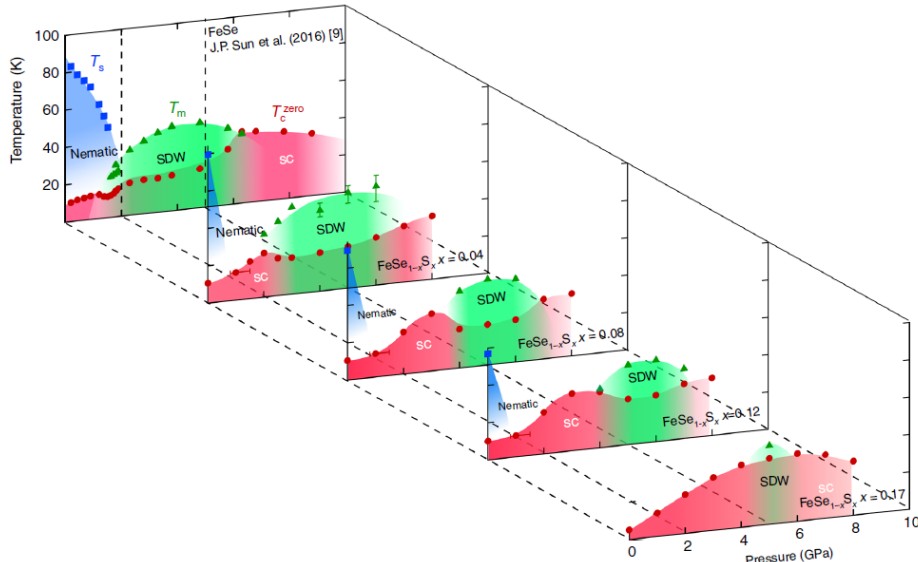

**Figure 8.** Temperature-pressure phase diagrams of $FeSe_{1-x}S_x$ (Figure reprinted from Matsuura, K. et al. *Nat. Commun.* **2017**, *8*, 1143. Copyright 2017 by Macmillan Publishers Limited).

## 3. Intercalated FeSe-Based Superconductors

In 2010, we discovered the alkali-intercalated FeSe-based superconductor with $T_c$ ~30 K for the first time [53,54]. Through Rietveld refinements against PXRD, an average structure of $K_{0.8}Fe_2Se_2$ (i.e., 122-type), was determined, which was iso-structural, to be known $BaFe_2As_2$, as seen in Figure 9. From the temperature dependence of resistance and magnetization, the $T_c{}^{onset}$ was identified as 30.1 K. The dominant carriers were n-type, suggesting an electron-doped superconductor. This was the first report of enhancing $T_c$ in an FeSe-based superconductor up to 30 K without applying high pressure. Following this discovery, several groups quickly identified many 122-type FeSe-based superconductors such as $Cs_{0.8}(FeSe_{0.98})_2$ ($T_c$ ~27 K) [55,56], $Rb_{0.88}Fe_{1.81}Se_2$ ($T_c$ = 32 K) [57], $Tl_{0.58}Rb_{0.42}Fe_{1.72}Se_2$ ($T_c$ ~32 K) [58], and (Tl, K)$Fe_xSe_2$ ($T_c$ ~31 K) [59]. A new family of materials thereby came into the awareness of the superconducting community.

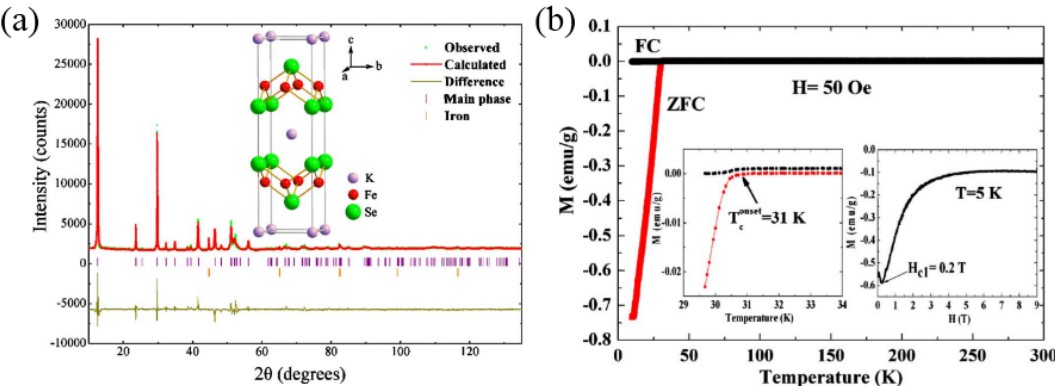

**Figure 9.** (**a**) PXRD pattern and Rietveld refinements profile of $K_{0.8}Fe_2Se_2$. Crystal structure is shown as inset. (**b**) Temperature dependence of magnetization of the $K_{0.8}Fe_2Se_2$ single crystal (Figure reprinted from Guo J.G. et al. *Phys. Rev. B* **2010**, *82*, 180520. Copyright 2010 by American Physical Society).

Transmission electron microscopy (TEM) on $K_{0.8}Fe_xSe_2$ and $KFe_xSe_2$ ($1.7 \leq x \leq 1.8$) samples revealed that complex phase-separation emerge in such $AFe_2Se_2$ superconductors. The superconducting phase is likely to be $K_xFe_2Se_2$, which was later confirmed by STM results. The Fe-vacancy ordered phase $K_2Fe_4Se_5$ showed a $\sqrt{5} \times \sqrt{5} \times 1$ superstructure [60]. The NPD on $K_xFe_2Se_2$ samples revealed that the superstructure phases $A_2Fe_4Se_5$ existed in all K, Cs, Rb, (Tl, Rb), and (Tl, K) intercalated FeSe-based

superconductors [61]. Li et al. proposed that the existence of Fe vacancies could locally destroy SC [62], and suggested that the $K_2Fe_4Se_5$ layer may be indispensable to regulate superconductivity in $KFe_2Se_2$ by providing charge carriers [63]. More superstructures associated with different Fe-vacancy and alkali-metal-ordering were later revealed later [64–67].

Phase separation hinders the investigation of the intrinsic property of FeSe-based 122-type superconductors. A low-temperature route, named the liquid-ammonia method, was proposed to synthesize phase-pure superconductors. A series of alkali and ammonia co-intercalated compounds were synthesized such as $Li(NH_3)_yFe_2Se_2$, $Na(NH_3)_yFe_2Se_2$, $Ca_{0.5}(NH_3)_yFe_2Se_2$, $Sr_{0.8}(NH_3)_yFe_2Se_2$, $Ba_{0.8}(NH_3)_yFe_2Se_2$, $Eu(NH_3)_yFe_2Se_2$, and $Yb(NH_3)_yFe_2Se_2$, with a $T_c$ of 30–46 K [68–70]. Taking $Na(NH_3)_yFe_2Se_2$ as an example, the crystal structure is composed of edge-sharing $FeSe_4$-tetrahedra layers and Na atoms between the $FeSe_4$ layers, and the $T_c$ $^{onset}$ is 46 K, as shown in Figure 10a,b. In addition, these authors built a discrete superconducting phase diagram against K content, as seen in Figure 10c,d, where the low intercalated content could enhance the $T_c$ to 44 K. After increasing the K content to 0.6, the $T_c$ was lowered to 30 K [71]. This trend is well reproduced by the density function theory and Monte Carlo simulations [72].

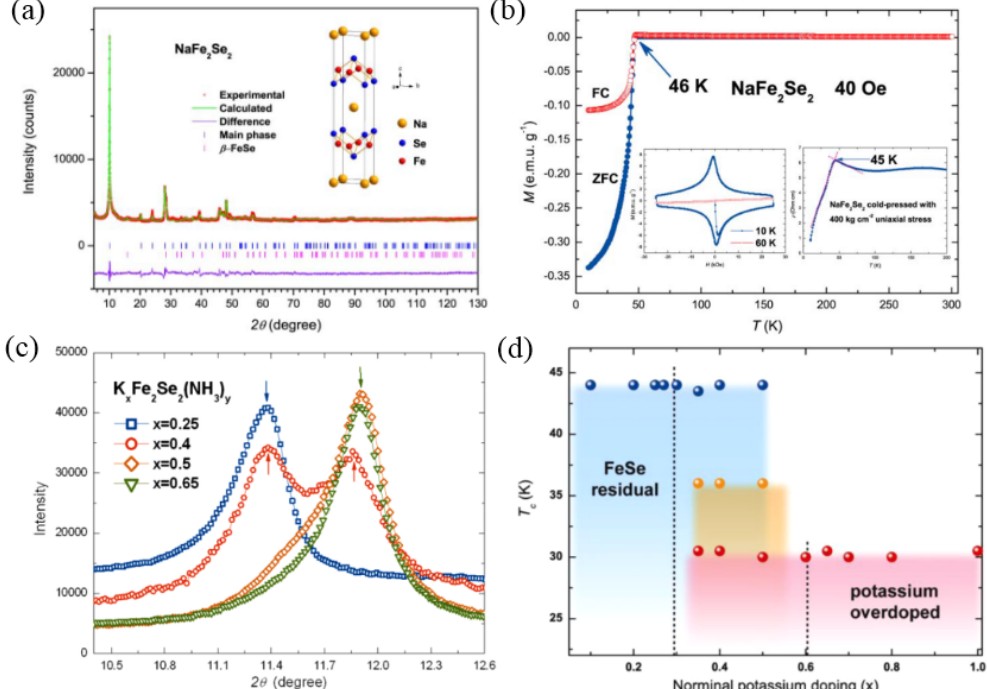

**Figure 10.** (**a**) PXRD pattern and Rietveld refinements profile of $NaFe_2Se_2$ at ambient temperature. Inset shows the crystal structure of $NaFe_2Se_2$. (**b**) Temperature dependence of magnetization of $NaFe_2Se_2$ polycrystalline sample. (**Left**) inset shows superconducting loops of $NaFe_2Se_2$ at 10 K and 60 K, respectively. (**Right**) inset shows temperature dependence of the electrical resistance of cold-pressed $NaFe_2Se_2$. (**c**) Enlargement of (002) peak of PXRD patterns for $K_xFe_2Se_2(NH_3)_y$ ($x$ = 0.25, 0.4, 0.5, and 0.65) samples measured at 297 K. (**d**) $T_c$s of $K_xFe_2Se_2(NH_3)_y$ as a function of nominal potassium content (Figure 10a,b reprinted from Ying, T.P. et al. *Sci. Rep.* **2012**, 2, 426. Copyright 2012 by Macmillan Publishers Limited. Figure 10c,d reprinted from Ying, T. P. et al. *J. Am. Chem. Soc.* **2013**, 135, 2951–2954. Copyright 2013 by American Chemical Society).

Later, the high-resolution NPD technique was adopted to determine the crystal structure of $Li_x(NH_3)_yFeSe$, where the Li and $NH_3$ are located in between the FeSe layer [73]. Jin et al. also determined the structure of $Na_{0.39}(C_2N_2H_8)_{0.77}Fe_2Se_2$ in this way, as shown in Figure 11 [74]. The existence of organic molecules seems not to be related to SC, but can drastically alter the geometry of FeSe, even changing the structural symmetry due to rotations of the long-chain of the C atom [75].

Lu et al. synthesized an intercalated superconductor $(Li_{0.8}Fe_{0.2})OHFeSe$ by the hydrothermal method. The $T_c$ was ~41 K and the precise structure parameters were determined by x-ray and NPD. the details of which are shown in Figure 12 [76]. The $T_c$ of different FeSe intercalates imply that $T_c$ is proportional to the interlayer spacing below 9.0 Å. Once the spacing is above 9.0 Å, $T_c$ seems to be independent on this spacing [77]. Therefore, the doped electron count is more important for determining $T_c$. Thus, studies of metal ions and molecular co-intercalated FeSe-based compounds are important to enhance $T_c$ and understand the effect of structural change on SC [78,79]. Additionally, under high pressure, a second superconducting phase with higher $T_c$ was observed in $K_xFe_2Se_2$, which can be ascribed to the quantum critical phase transition [80,81]. Such pressure-induced SC was also verified in $(Li_{0.8}Fe_{0.2})OHFeSe$ [82] and in $Li_x(NH_3)_yFe_2Se_2$ [83].

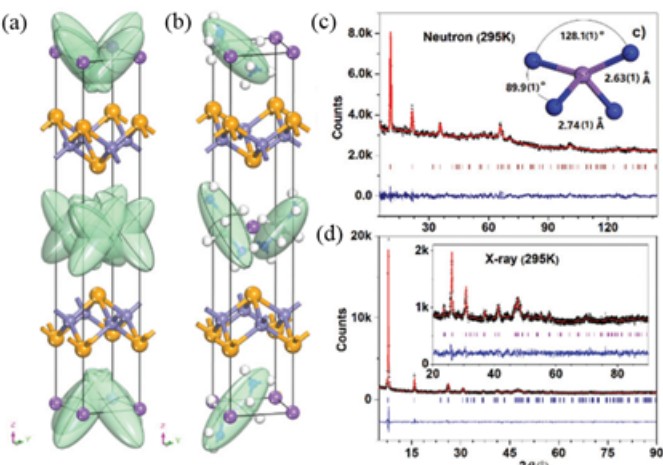

**Figure 11.** (**a–b**) Crystal structure of $Na_{0.39}(C_2N_2H_8)_{0.77}Fe_2Se_2$. (**c**) Rietveld refinements of NPD pattern. Inset is the atomic geometry of $FeSe_4$ tetrahedra. (**d**) Rietveld refinements of PXRD pattern. Inset shows the expanded range of 20°–90°. [Figure reprinted from Jin S.F. et al. *Chem. Commun. (Cambridge, U. K.)* **2017**, 53, 9729-9732. Copyright 2017 by The Royal Society of Chemistry.].

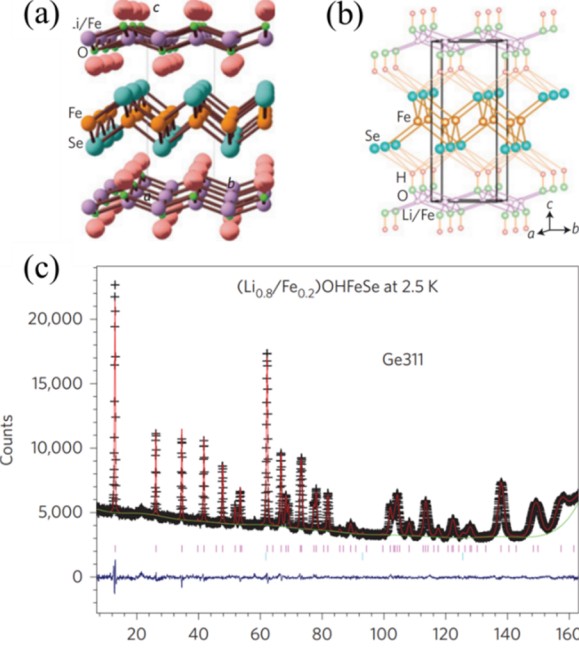

**Figure 12.** (**a**,**b**) Crystal structure of $(Li_{0.8}Fe_{0.2})OHFeSe$. (**c**) Rietveld refinements of PND pattern of $(Li_{0.8}Fe_{0.2})OHFeSe$ (Figure reprinted from Lu X.F. et al. *Nat. Mater.* **2015**, 14, 325–329. Copyright 2015 by Macmillan Publishers Limited).

Sizeable single-crystals of $A_x Fe_2 Se_2$ and $(Li_{0.8}Fe_{0.2})OHFeSe$ superconductors can be easily grown by the self-flux and hydrothermal method. Thus, the important Fermi surface topology can be detected by ARPES experiments. The most important finding is that the hole-pocket at the $\Gamma$ point sinks tens of meV below the Fermi level and the unique Fermi surface only contains electronic pockets [84]. In terms of the approximation of a rigid band shift, it is a consequence of electron-doping, which lifts up the Fermi energy. This topology of the Fermi surface has also been observed in $(Li_{0.8}Fe_{0.2})OHFeSe$ (see Figure 13) [85,86]. Such a Fermi surface challenges the established picture of electron-hole nesting for inducing SC in FeAs-based superconductors [9,10,87]. At the same time, the ARPES measurements confirmed that the superconducting gap was of the isotropic type, with a magnitude of 10.3 meV. This nodeless gap suggests that a conventional *s*-wave pairing could better describe the origin of SC [88].

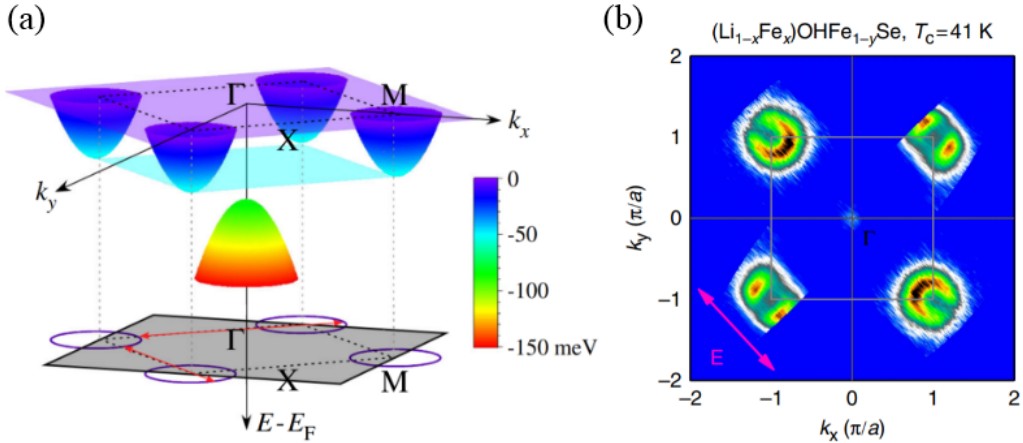

**Figure 13.** (**a**) Electronic band structure of $K_{0.8}Fe_{1.7}Se_2$ and possible $(\pi, \pi)$ scattering vector. (**b**) Fermi surface of $(Li_{0.84}Fe_{0.16})OHFe_{0.98}Se$ measured at 20 K (Figure reprinted from Qian T. et al. *Phys. Rev. Lett.* **2011**, 106, 187001. Copyright 2011 by American Physical Society. Reprinted from Zhao L. et al. *Nat. Commun.* **2016**, 7, 10608. Copyright 2016 by Macmillan Publishers Limited).

## 4. Superconductivity of FeSe Film

In 2012, Wang et al. reported that a single unit cell (SUC) FeSe film grown on a $SrTiO_3$ (STO) substrate by molecular beam epitaxy (MBE) exhibited a very large superconducting gap ($\Delta$) of 20 meV. Assuming that the superconducting mechanism of both FeSe film and bulk FeSe is the same, the estimated $T_c$ of the SUC FeSe film may be larger than 77 K [89]. The SUC FeSe film had an atomically smooth surface after Se-flux treatment. The temperature-dependent resistance showed a little lower $T_c^{onset}$ of 53 K as plotted in Figure 14a. The upper inset in Figure 14a shows that the SC is slowly suppressed by the external magnetic field. The tunneling spectrum in the SUC FeSe film clearly exhibited a ~20 meV superconducting gap, as seen in Figure 14b. However, the double unit cell (DUC) FeSe film did not exhibit SC at all (see Figure 14c). It seems that the interfacial effect offers an effective way to realize high $T_c$ in FeSe. Ge et al. reported a rather high $T_c$, above 100 K, in FeSe/STO film by using in situ four-point probe electrical transport measurements. This $T_c$ is the highest value in all reported Fe-based superconductors, which is a rather exciting result that deserves further investigation [90]. Subsequently, many FeSe thin films grown on different substrates have been studied in order to understand the mechanism of enhancing SC.

He et al. studied the topology of the Fermi surface of the SUC FeSe film grown on a STO substrate by changing the carrier concentration through a different annealing procedure [91]. Surprisingly, they found that two competitive phases, a non-superconducting phase at low-doping level and another superconducting phase at high-doping level, appeared during the annealing process. The electronic structure and the superconducting gap were tuned in a larger range by adjusting the annealing time and temperature. As depicted in Figure 15a–d, it was found that the magnitude of the superconducting

gap increased from ~10 meV to ~19 meV. Meanwhile, the relation between the superconducting gap size and temperature could be well described by the BCS theory, which is displayed as a green line in Figure 15e–h). Meanwhile, Tan et al. also grew a high-quality thin FeSe film on a STO substrate and measured the Fermi surface and band dispersion [92]. They found that a short-range SDW appeared in the thin FeSe film (50 unit-cell). As depicted in Figure 16a, the SC arises while the spin density wave is suppressed. This phase diagram reveals that there is a unified trend in SUC FeSe/STO and other Fe-based superconductors by controlling the doping level and lattice constants. It also implies that a higher $T_c$ will appear in the FeSe film if the lattice constants are expanded and higher electronic doping is introduced. Rebec et al. compared the $T_c$ and the scale of superconducting gap in 60 unit-cell FeSe/STO (001), SUC FeSe/STO (001), and 3 UC FeSe/STO (001) coated by potassium atoms and SUC FeSe/TiO$_2$ (100). By analyzing the ARPES results, they found that electron doping plays an important role in inducing high $T_c$ [93].

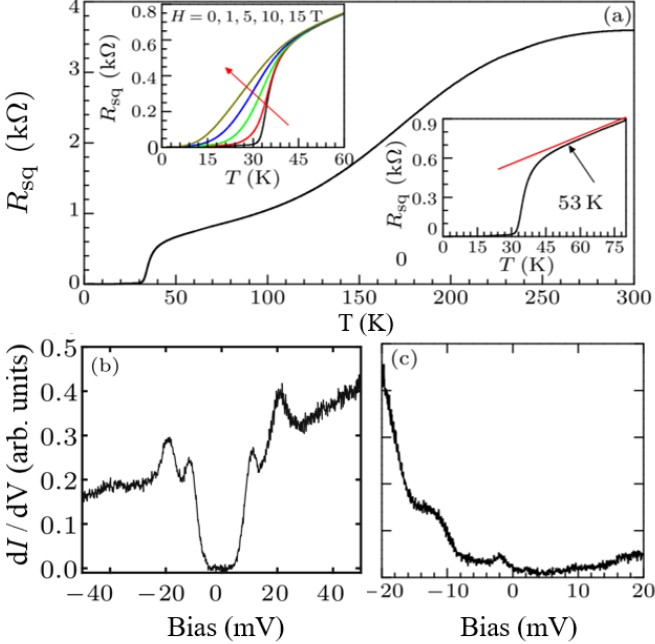

**Figure 14.** (**a**) Temperature dependence of square resistivity of a 5-UC-thick FeSe film on insulating STO (001). (**b**) Tunneling spectrum measurement of the 1-UC-thick FeSe film on STO (001) at 4.2 K. (**c**) dI/dV spectrum of the 2-UC-thick FeSe film on insulating STO (001) at 4.2 K. (Figure reprinted from Wang, Q.Y. et al. *Chin. Phys. Lett.* **2012**, *29*, 037402. Copyright 2012 by Chinese Physical Society and IOP Publishing Ltd.).

There is an interaction between electrons from FeSe and the oxygen optical phonons of the STO substrate, thus enhancing the electron-phonon coupling [94]. Figure 17 suggests that there is a close relationship between interfacial electron–phonon coupling and $T_c$ enhancement. Both experimental and calculation results revealed that interfacial electron–phonon coupling could account for the enhancement of $T_c$ in the SUC FeSe/STO film. To find out the main factor in enhancing SC, Ding et al. grew SUC FeSe films on a TiO$_2$ (001) substrate [95]. As depicted in Figure 18, a TiO$_2$ film with a thickness of 15 nm was grown on a STO substrate. On the TiO$_2$ surface, a $4 \times 1$ reconstruction of the oxygen vacancies (white spots) can clearly be seen. The density of the oxygen vacancies in the as-grown TiO$_2$ surface was about $4.6 \times 10^{-2}$ per nm$^2$, and post-annealing reduced the vacancies of TiO$_2$ to $6.1 \times 10^{-3}$ per nm$^2$, as shown in Figure 18a–c. The SUC FeSe film on TiO$_2$ also exhibited a clear superconducting feature in the dI/dV spectrum, but the double unit cell (DUC) FeSe was not superconducting (see Figure 18d). In the SUC FeSe sample grown on a TiO$_2$ substrate with different oxygen vacancies, there was no change in the superconducting gap (see Figure 18e). Thus, the density of interfacial oxygen vacancies has a limited effect on enhancing the interfacial charge transfer and $T_c$.

Zhou et al. also grew SUC FeSe films on a MgO (001) substrate, and the onset $T_c$ was 18 K, as inferred by transport measurement. [96] The atomic image illustrates that the SUC FeSe film grown on the MgO (001) substrate was along FeSe (001). A 1.3% in-plane tensile strain was induced by the lattice mismatch of the FeSe and the MgO (001) substrate. Scanning transmission electron microscopy (STEM) images of the interfacial structure demonstrated that Mg atoms could be replaced by Fe atoms, and the result of the calculations showed that the MgO film near the interface becomes electron doped, which shows the capability of enhancing the charge-transfer effect.

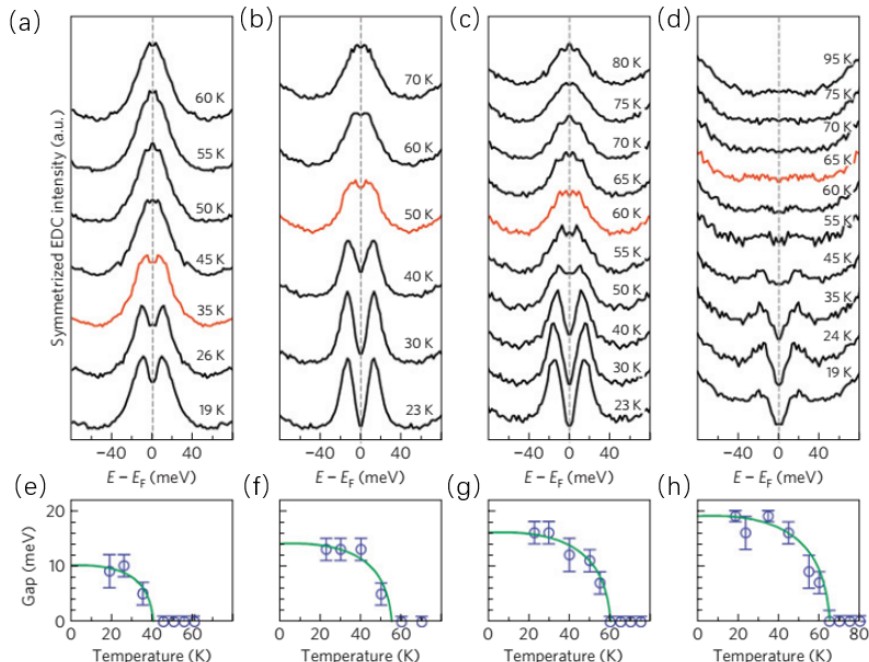

**Figure 15.** (**a**–**d**) Symmetrized energy distribution curves (EDCs) of the Fermi surface near the M point in the Brillouin zone with various annealing process. (**e**–**h**) Temperature dependence of the superconducting gap. Green lines are fitting curves based on the BCS model (Figure reprinted from He, S.L. et al. *Nat. Mater.* **2013**, 12, 605–610. Copyright 2013 by Macmillan Publishers Limited.).

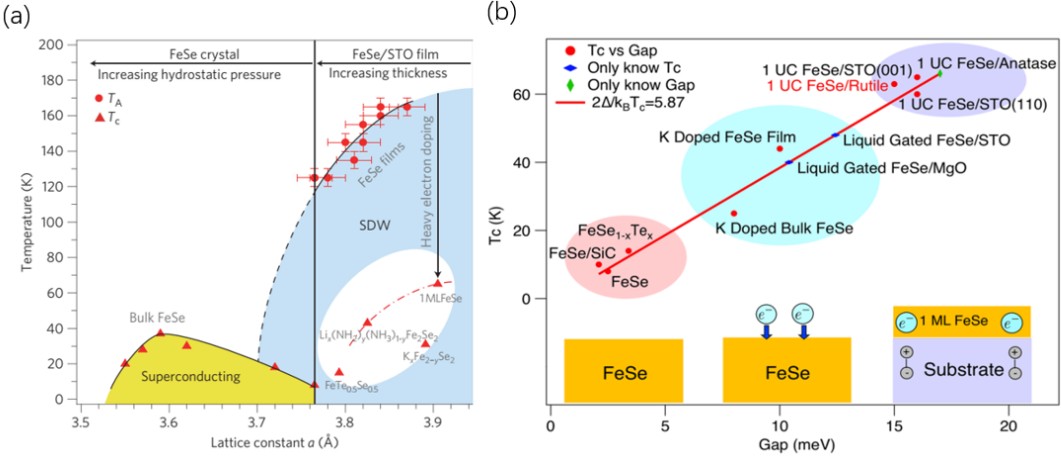

**Figure 16.** (**a**) The proposed phase diagram of various FeSe film and bulk FeSe. (**b**) $T_c$ and superconducting gap of bulk FeSe, FeSe grown on the SiC (0001) substrate, K-doped thin FeSe film, K-doped FeSe film grown on the STO (001) substrate, single unit cell (SUC) FeSe/STO (001), and FeSe/STO (110). (Figure 16a reprinted from Tan, S.Y. et al. *Nat. Mater.* **2013**, 12, 634–640. Copyright 2013 by Macmillan Publishers Limited. Figure 16b reprinted from Rebec, S.N. et al. Phys. Rev. Lett. 2017, 118, 067002. Copyright 2017 by American Physical Society.).

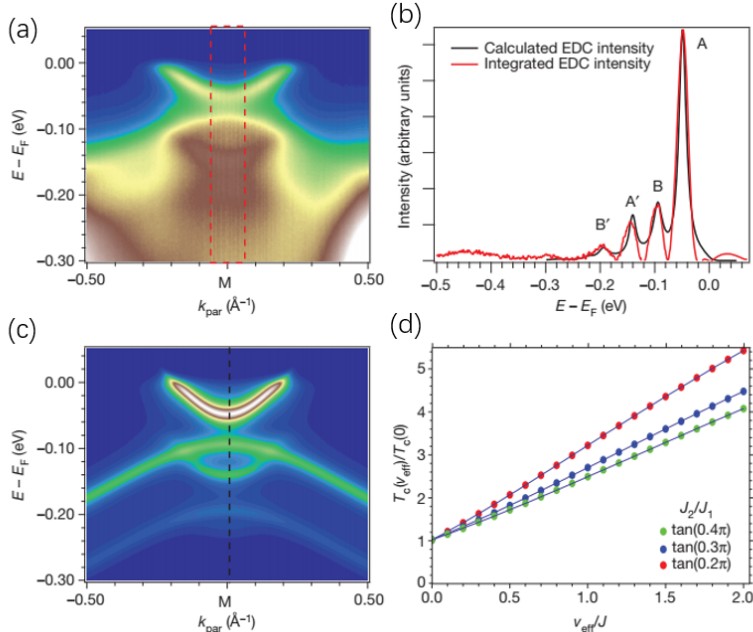

**Figure 17.** Electron-phonon coupling effect and measurement of $T_c$ enhancement. (**a**) High-statistics scan at the M point of the first Brillouin zone measured at 10 K. (**b**) The EDC after background subtraction and the calculated counterpart. (**c**) Theoretical calculation involving hole and electron bands coupled to a dispersion phonon mode via model spectral functions, and the dashed line is the symmetrized energy distribution curves. (**d**) $T_c$ enhancement as a function of electron–phonon coupling strength ($V_{eff}/J$). (Figure 17 reprinted from Lee, J.J. et al. *Nature* **2014**, *515*, 245–248. Copyright 2014 by Macmillan Publishers Limited).

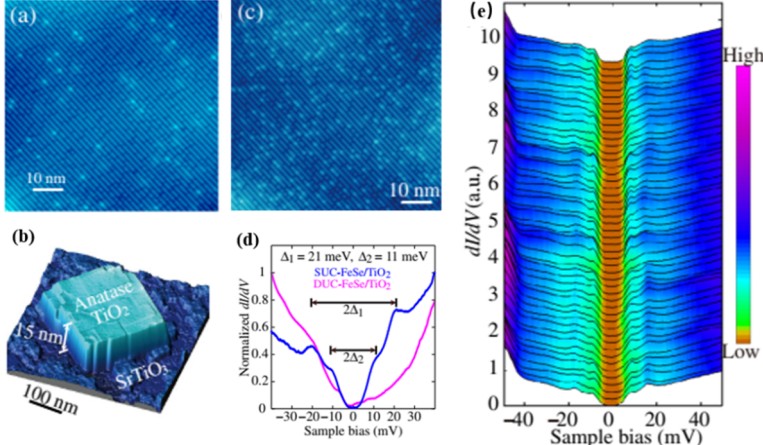

**Figure 18.** (**a**) STM topography of $TiO_2$ with lower density of oxygen vacancies after annealing (bright spots). (**b**) STM topography in a $TiO_2$ island supported by the STO substrate. (**c**) Enlarged STM topography acquired on a $TiO_2$ island with oxygen vacancies (without annealing). (**d**) Low energy $dI/dV$ spectra taken on the SUC and DUC FeSe films. (**e**) A series of $dI/dV$ spectra acquired from the FeSe surface along one direction (Figure 18 reprinted from Ding, H. et al. *Phys. Rev. Lett.* **2016**, *117*. Copyright 2016 by American Physical Society).

Miyata et al. observed SC with a $T_c$ of 48 K in a multilayer FeSe film by coating potassium onto its surface [97]. The electronic structure of the SUC FeSe films are shown in Figure 19. The Fermi surface of the SUC FeSe film consists of a large electron pocket at the M point of the Brillouin zone, which is consistent with those of $K_x Fe_2 Se_2$ and LiOHFeSe. However, the Fermi surface of the triple unit-cell FeSe film included a hole pocket at the Γ point and a tiny electron pocket near the M point of the

Brillouin zone (see Figure 19d–f), which is similar to that of bulk FeSe. After coating more K atoms, a downward shift of hole-like band occurred at the Γ point. The hole-pocket at the Γ point thereby gradually disappeared, as shown in Figure 19g–o. The total doped-electron count was about $0.11e^-$ in the triple unit-cell FeSe film, and a superconducting gap of 8 meV opened at $T_c$ = 46 K and closed above 51 K. Obviously, electron doping by coating K atoms is a useful strategy to induce high $T_c$ in multilayer FeSe films.

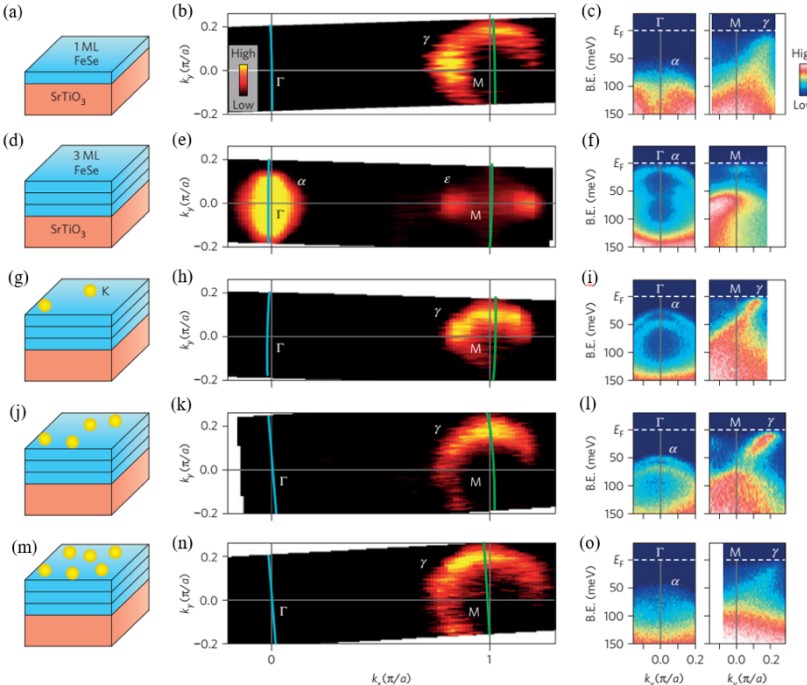

**Figure 19.** (**a,d,g,j,m**) Schematic diagram of the FeSe films with different thicknesses and content grown on the STO substrate. (**b,e,h,k,n**) Fermi surfaces mapped out by ARPES. Green and blue lines are momentum cuts around the Γ and M points. (**c,f,i,l,o**) ARPES intensity image near the Fermi energy ($E_F$) as a function of binding energy and wav-vector along the cuts close to Γ and M. (Figure 19 reprinted from Miyata, Y. et al. *Nat. Mater.* **2015**, *14*, 775–779. Copyright 2015 by Macmillan Publishers Limited).

Interfacial electron–phonon coupling has also been proposed to explain the enhancement of high $T_c$ in SUC FeSe/STO. However, the evidence that can directly connect interfacial electron–phonon coupling intensity (EPI) with high $T_c$ is still not sufficiently clear. Song et al. studied thin films of FeSe by replacing $^{16}$O with its isotope $^{18}$O [98]. Surprisingly, they found that the energy difference between the electron-type band and its replica band was approximately in proportion to the inverse square root of the mass of oxygen atoms. Therefore, the SC is highly related to interfacial EPI. As depicted in Figure 20, superconducting gaps $\Delta_1$ and $\Delta_2$ were ~9.5 meV while the ratio η (i.e., the EPI) is close to zero, which is basically equal to the superconducting gap of the K-dosed multilayer FeSe film. Thus, it demonstrated that a synergistic effect between possible spin fluctuations and enhanced EPI results in such high $T_c$ in FeSe/STO. It is worth noting that theorists could qualitatively describe the Fermi surface, hidden magnetic order, replica bands, and the superconductivity isotropic Cooper pairs in heavily electron-doped FeSe by an extended Hubbard model [99,100]. Such explanations could deepen the understanding of the superconducting mechanism and may help us explore more high $T_c$ superconductors.

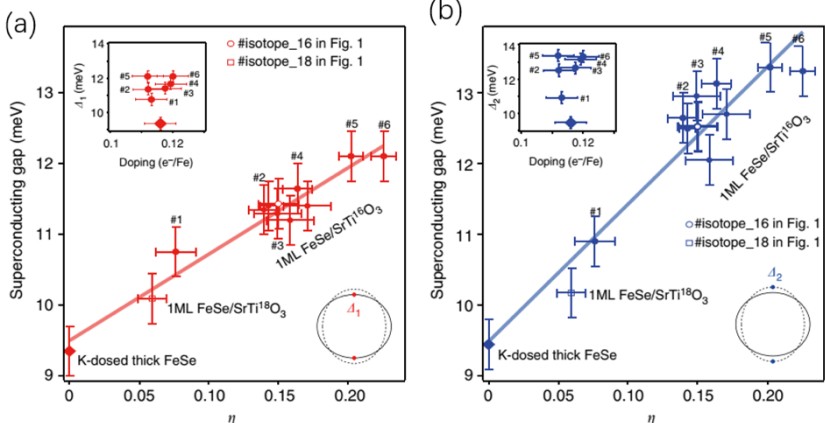

**Figure 20.** (**a**,**b**) Relationship between the superconducting gap $\Delta_1$, $\Delta_2$ and the intensity ratio ($\eta$) between the side band and the main band. The insets show the connection between the superconducting gap sizes and doping for corresponding samples. (Figure 20 reprinted from Song, Q. et al. *Nat. Commun.* **2019**, 10, 758. Copyright 2019 by Macmillan Publishers Limited).

## 5. Conclusions

The finding of a series of surprisingly high $T_c$ in FeSe-based materials is a significant breakthrough in the superconducting community and sets up a very interesting superconducting family. However, the microscopic mechanism of the enhancement of $T_c$ through multiple treatments is still elusive. Many kinds of effects have been proposed to explain the enhancement in $T_c$ such as coupling between the electrons and phonons of the substrate, the tensile strain effect introduced by lattice mismatch between FeSe film and the substrate, and the charge transfer, *etc*. More subtle experiments should be conducted to figure out which effect plays the decisive role. In addition, as FeSe thin films are very thin and are easily oxidized, the transport and magnetic properties are hard to measure. Therefore, new methods and more novel experimental tools should be developed to detect the intrinsic physical properties.

**Author Contributions:** Investigation, K.Z., J.W.; Y.S. writing—original draft preparation, K.Z.; J.W.; L.G.; J.-g.G.; writing—review and editing, K.Z.; J.W.; J.-g.G.; funding acquisition, J.-g.G.

**Funding:** This research was funded by the National Natural Science Foundation of China, grant number 51922105 and 51772322, the National Key Research and Development Program of China, grant number 2016YFA0300600 and 2017YFA0304700, and the Chinese Academy of Sciences, grant number QYZDJ-SSW-SLH013.

**Conflicts of Interest:** The authors declare no conflict of interest.

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
