# Peer review of "Highly-Tunable Crystal Structure and Physical Properties in FeSe-Based Superconductors"

_crystals, doi:10.3390/cryst9110560_

Round 1

Reviewer 1 Report

The authors review the superconductor FeSe as well as electron-doped FeSe superconductors.  The english in the manuscript is unfortunately very bad.  Below is a list of corrections. I strongly recommend that a native english speaker revise the language in the manuscript!

Also, the authors should cite

Hirschfeld, Korshunov, Mazin, Rep. Prog. Phys. 74, 124508 (2011) and

Chubukov, "Iron-Based Superconductivity", Series in Materials Science, 2015; v. 211, p. 255

when they mention the inapplicability of the Fermi-surface-nesting scenario for superconductivity in iron superconductors in lines 51, 129, 208, and 293.  By contrast, it is worth mentioning that recent theoretical work at the strong correlation limit shows promise in describing the electronic structure and superconductivity in electron-doped FeSe:

Rodriguez, Phys. Rev. B 95, 134511 (2017)

Rodriguez and Melendrez, J. Phys. Communications 2, 105011 (2018).

Also, the authors state at various points in the manuscript that "four" electron pockets exist at the M point.  In fact, only two do! See, for example, the inset in Fig. 20.

List of Corrections:

line     proposed change or question

10   "property" -> "properties" and "have been" -> "are"

11   "against" -> "versus"

12   "site exhibits" -> "sites exhibit" and "exertion" -> "application"

16   "At last" -> "Last" , "from" -> "shown by",

       and "introduced" -> "discussed"

17   "and" -> "and the"

18   "step" -> "step-wise" and "indicates" -> "indicates how"

27   "brings the" -> "brought" , "into a fresh" -> "to fresh new" , and

       "binary FeSe" -> "binary material"

34   "lacked" -> "lacking"

38   "inaccessible" -> "hence inaccessible"

40   "the technique" -> "a technique" and "overcame" -> "found"

41   "reveal" -> revealed"

42   "important" -> "importantly"

43   "with" -> "with a"

44   "for" -> "in the"

47   "superconductor" -> "superconductivity"

49   "four" -> "two" or remove "four" 

50   "unique" -> "a unique" and "Copper" -> "Cooper"

56   "part" -> "part of the paper"

60   "it" -> "this review"

61   "explore" -> "motivate the exploration of"

62   "of" -> "in"

64   ", which is regard" -> ".  It is regarded"

66   "layer that is" -> "layers that are"

67   "large" -> "a large"

68   "gives out" -> "yield"

69   remove "temperature", and references?

81   references?

91   "Besides" -> "Also" and "what" -> "which"

99   references?

100 "parent" -> "parent compound"

102 remove "could"

103 "About" -> "Concerning"

104 "But a" -> "Only"

107 "grow" -> "grew"

112 "that the" -> "a"

113 "is" -> "of" and "by" -> "by the"

114 "Hall" -> "the Hall"

115 remove "But"

116 "shown" -> "however, as shown"

120 "glues" -> ""glues""

121 "temperature-dependent" -> "a temperature-dependent" and

      "and fitted it" -> "which they fit"

122 "BCS" -> "by the standard BCS"

123 "gap and the" -> "gap. The"

124 "two-gap" -> "a two-gap"

129 "Fermi nesting -> "Fermi surface nesting"

136 "relates" -> "is related"  and "low temperature" -> "low-temperature"

137 "favor to be an" -> "possibly due to"

138 "of" -> "of an"

139 "with its" -> "with"

140 "gives out" -> "suggests"

164 "Application" -> "The application" and "exhibits" -> "exhibits the"

165 "FeSe polycrystalline" -> "a polycrystalline FeSe"

167 "applied a" -> "applied"

168 "to the" -> "to"

170 "of" -> "in"

181 "contrast" -> "contrary"

187 "nature of" -> "between"

188 "see Figure 6" -> "(see Figure 6)" and

      "linear resistivity against temperature" -> "linear-T resistivity"

189 "can" -> "can be" and "of" -> "of a"

191 "what" -> "which"

195 "enhanced" -> "enhance"

197 "convinced" -> "convincing"

198 "grew" -> "grown" and "of -> "of the"

203 "that the" -> "a" and "is" -> "that is"

205 "SC and structure" -> "the SC and the structure"

212 "Besides" -> "Also"

215 "significant" -> "significantly" and "BCS" -> "the BCS"

217 "and" -> "bands and"

220 "increasing S-content" -> "the S-content increased"

221 not clear: "pressure-induced was gradually narrowed"

222 "will become" -> "becomes"

246 "temperature" -> "the temperature"

247 "dominated" -> "dominant" and "is" -> "was"

248 "of" -> "in an"

251 "It demonstrates" -> "This demonstrated" and "starts to walk" -> "came"

252 "vision of superconducting" -> "view of the superconductivity"

262 "disadvantaged to study" -> "problematic when studying"

264 "superconductor" -> "superconductors"

268 "Besides, they" -> "Also, (ref.)"

271 "Mont" -> "Monte" , and references?

291 "rigid model of band shifting" -> "the approximation of a rigid band shift"

      and "uplifts" -> "lifts up"

293 "Such" -> "Such a" and "for inducing" -> "inducing"

295 references?

296 "even" -> "even if"

297 "still" -> "remains"

326 "thinned" -> "atomically thin"

329 "Temperature-dependent" -> "The temperature-dependent"

334 "Then" -> "Subsequently"

335 "to" -> "in order to"

337 "excited" -> "exciting"

355 "and" -> "and the"

356 "was" -> "were"

357 "and" -> "and the"

360 "as" -> "by the"

362 "that" -> "that a"

364 "restrained -> "suppressed"

368 "are" -> "is" and "scale" -> "the scale"

390 references?

399 "have" -> "have a"

426 "gardully" -> "gradually"

436 "connection" -> "the connection"

442 "that" -> "that a"

443 "of" -> "between"

444 "SC" -> "the SC"

446 "elusive, many" -> "elusive.  Many"

450 "Besides, in the" -> "Also, because" and

      "thin film, it is too thin" -> "films are very thin"

451 "may not" -> "cannot" and "or" -> "and"

452 "for detecting the" -> "to detect" , "property" -> "properties" and

      "Of course, it sill need" -> "Clearly"

453 "work" -> "will be needed to work"

Reviewer 2 Report

The authors present a review article on a variety of the iron chalcogenide superconductors.  The main text itself seems to be well written, but there are no citations in the main text, while publications are listed in the reference section.  The authors should correct this.

As for other points,

-On line 122, the authors wrote The isotropic gap features unambiguously exclude the existence of nodes in the superconducting gap, But later the author also described that FeSe has an extremely anisotropic gap (on line 138 (ref 26?, 27?) in the manuscript).  I think the authors should explain the reason for this discrepancy. 

-On line 200, the authors described the annealing effect. There is a lot of research on the annealing effect (see Y. Sun et al., Supercond. Sci. Technol 32 (2019) 103001 and reference therein).  I do not know what literature the authors intended to cite here, but at least, citing only ref. 41 is not appropriate.

-On line 327, the authors wrote, In 2012, Wang et al. reported that single unit cell (SUC) FeSe film grown on SrTiO3 (STO) substrate by molecular beam epitaxy (MBE) exhibited very high Tc (~65 K).  However, in the paper Wang et al., Chin. Phys. Lett. 29 (2012) 037402 (ref 84), the Tc value of 65 K was not reported.  The author should cite appropriate references.

-In Fig. 14, the authors showed a dI/dV spectrum of Wang et al., Chin. Phys. Lett. 29 (2012) 037402. The spectrum data was obtained in a special sample  that was grown in a different condition from that of their best samples for resistivity measurement. Why did the authors not show the best data of Wang et al., which shows much larger gap of 20 meV?

-In line 362, the authors wrote They found that short-range spin density wave (SDW) appeared in thin FeSe film (50 unit-cell). I think the observation of SDW is contradictory to the results in bulk samples. Can the authors provide an explanation?

Round 2

Reviewer 1 Report

See the attached pdf file for corrections to english.

Author Response

Dear Reviewer,

    I appreciate your patience for improving our manuscript. All suggestions and comments have been corrected in the revised version. Let me know if you have any further questions.

    Best,

    Jian-gang Guo 

Reviewer 2 Report

The issues that I previously pointed out were clearly explained in this revised version of the manuscript. I think the manuscript would be appropriate for publication after the authors address the following minor issues.

Several important citations in the previous version of the manuscript were somehow removed in the present version. (For example, ref1-8 in the previous version, including [Kamihara et al JACS 2008] and [Hsu et al., PNAS 2008]) The authors should confirm it. The citation for ref.6-92 in the main text were numbered in Roman numerals. The authors should refer the style guide of the journal and unify the notation.

Author Response

Dear Reviewer,

    Thanks for you comments. The issues about references come from the post-editing of the editor. Our revised manuscript already includes all the necessary references you mentioned.

    Best wishes,

    Jian-gang Guo